# Design, Synthesis and Characterization of [G10a]-Temporin SHa Dendrimers as Dual Inhibitors of Cancer and Pathogenic Microbes

**DOI:** 10.3390/biom12060770

**Published:** 2022-05-31

**Authors:** Arif Iftikhar Khan, Shahzad Nazir, Aaqib Ullah, Muhammad Nadeem ul Haque, Rukesh Maharjan, Shabana U. Simjee, Hamza Olleik, Elise Courvoisier-Dezord, Marc Maresca, Farzana Shaheen

**Affiliations:** 1Third World Center for Science and Technology, International Center for Chemical and Biological Sciences, University of Karachi, Karachi 75270, Pakistan; khanformanite@gmail.com (A.I.K.); nazirshahzad39@gmail.com (S.N.); aqibafridi209@gmail.com (A.U.); nadeem.and.chem@gmail.com (M.N.u.H.); 2H. E. J. Research Institute of Chemistry, International Center for Chemical and Biological Sciences, University of Karachi, Karachi 75270, Pakistan; rukeshmaharjan2013@gmail.com (R.M.); shabana.simjee@iccs.edu (S.U.S.); 3Dr. Panjwani Center for Molecular Medicine and Drug Research, International Center for Chemical and Biological Sciences, University of Karachi, Karachi 75270, Pakistan; 4Aix-Marseille Univ., CNRS, Centrale Marseille, iSm2, 13013 Marseille, France; hamza.olleik@live.com (H.O.); elise.courvoisier-dezord@univ-amu.fr (E.C.-D.)

**Keywords:** temporin SHa, D-alanine, antimicrobial peptide, Rink amide resin, dendrimer, antibacterial, anticancer

## Abstract

As the technologies for peptide synthesis and development continue to mature, antimicrobial peptides (AMPs) are being widely studied as significant contributors in medicinal chemistry research. Furthermore, the advancement in the synthesis of dendrimers’ design makes dendrimers wonderful nanostructures with distinguishing properties. This study foregrounds a temporin SHa analog, [G10a]-SHa, and its dendrimers as globular macromolecules possessing anticancer and antibacterial activities. These architectures of temporin SHa, named as [G10a]-SHa, its dendrimeric analogs [G10a]_2_-SHa and [G10a]_3_-SHa, and [G10a]_2_-SHa conjugated with a polymer molecule, i.e., Jeff-[G10a]_2_-SHa, were synthesized, purified on RP-HPLC and UPLC and fully characterized by mass, NMR spectroscopic techniques, circular dichroism, ultraviolet, infrared, dynamic light scattering, and atomic force microscopic studies. In pH- and temperature-dependent studies, all of the peptide dendrimers were found to be stable in the temperature range up to 40–60 °C and pH values in the range of 6–12. Biological-activity studies showed these peptide dendrimers possessed improved antibacterial activity against different strains of both Gram-positive and Gram-negative strains. Together, these dendrimers also possessed potent selective antiproliferative activity against human cancer cells originating from different organs (breast, lung, prostate, pancreas, and liver). The high hemolytic activity of [G10a]_2_-SHa and [G10a]_3_-SHa dendrimers, however, limits their use for topical treatment, such as in the case of skin infection. On the contrary, the antibacterial and anticancer activities of Jeff-[G10a]_2_-SHa, associated with its low hemolytic action, make it potentially suitable for systemic treatment.

## 1. Introduction

In the past few decades, with the continuous advancement in the technology for peptide development, the potential of the peptides has been thoroughly exploited. Peptide-based therapeutics have played a pivotal role in medical practice and thus has emerged as a promising tool to cure human diseases including tumors and multidrug resistant pathogen-associated infections [1]. It prompted better treatment results with targeted action on the overexpressed membrane receptors of diseased tissues and minimized side effects [2]. Moreover, their better penetration, lower immunogenicity as well as ease of synthesis improve their targeting role by many folds [3]. In addition, partial or complete D-amino-acid substitution; de-novo design by not utilizing any template sequence; side-directed modifications with the addition, deletion and substitution of amino acids; and self-assembled based architectures such as dendrimers, are reported as highly useful from the perspective of developing improved therapeutic activities, enhancing the resistance to proteolysis and surmounting the myriad of obstructions faced by host defense peptides [4]. Dendrimers, owing to their unique chemical characteristics and structure, have offered myriad of opportunities to researchers in medicinal chemistry. They possess a diversity of potential building blocks and their three dimensional fashion, along with innumerable possibility in design, lead to a globular architecture that mimics the globular proteins. Additionally, the relatively unexplored area of dendrimers makes them sought-after candidates for various biological applications [5]. As compared to polymers, dendrimers are monodispersed and well-defined in size. Due to these traits, these multivalent dendrimers have improved shelf lives, restrict ingress to their proteolytic sites and enhance the local concentration of peptide units to the membranolytic pursuit [4]. In addition, their various analogs have been employed as carriers of drugs, thus enhancing the solubility and biological activity of macromolecules. The multifunctionalization of peptides is due to the decorated surface of a dendrimer, with peptides that enhance their activity manifold as compared to peptides themselves. The dendrimer of gonadotropin-releasing hormone (GnRH) against prostate cancer is a well-known example [6,7].

Keeping in view all these traits of dendrimers, and as an initiative aimed at targeting cancer and microbes and addressing the needs of new and more selective anticancer and antimicrobial agents, we applied the concept of antimicrobial peptides to develop new dendrimeric peptides [8]. Temporin-SHa, segregated from the skin of Sahara frog *Pelophylax saharicus*, was identified as a potent inhibitor of Gram-positive and Gram-negative bacteria [9,10], as well as yeasts, fungi, and protozoa, along with anti-parasitic activity against promastigote and the intracellular stage (amastigote) of *Leishmania infantum* J [11,12,13]. Our group had previously synthesized potent anti-cancer linear analogs of temporin-SHa. These analogs manifested antiproliferative action against cancerous cells [14]. Potency against microbes, especially methicillin resistant *Staphylococcus aureus* (MRSA) [15] and *Helicobacter pylori* [16] was also determined. Based on our earlier studies, we developed dendrimeric analogs of previously identified potent analog [G10a]-SHa. These dendrimers were generated around lysine residue to formulate dimer and trimer or to conjugate [G10a]-SHa with a trimeric polymer core of a chemically modified Jeffamine linker to evaluate the utility of the Jeffamine-based [G10a]_2_-SHa conjugate. This trimeric Jeffamine polymer was earlier utilized in the discovery of anticancer peptidomimetics from one-bead-one-compound (OBOC) combinatorial libraries [17]. Herein, we report the synthesis, characterization, and anticancer as well as antimicrobial activity of dendrimers of the temporin-SHa analog [G10a]-SHa, which include dimer [G10a]_2_-SHa, trimer [G10a]_3_-SHa as well as a conjugation of dimer with the Jeffamine linker to formulate the conjugate named as Jeff-[G10a]_2_-SHa (Figure 1 and Table 1). The structural features of peptide dendrimers such as size, morphology, zeta potential, as well as antimicrobial and anticancer efficacy were also evaluated.

[G10a]-SHa is D-alanine substituted analog of temporin SHa having antibacterial [15,16] and anticancer [14] properties. It has a length of 13 residues and an amidated C-terminus. This peptide has one free amino group at the *N*-terminus and hydrophobic residues. It is cationic amphipathic and has alpha-helical moiety (net positive charge of +2 at pH 7). [G10a]_2_-SHa has a length of 27 residues and an amidated C-terminus. This dimer has two free amino groups at the *N*-terminus and hydrophobic residues. It is cationic amphipathic and has alpha-helical moiety containing D-alanine (net charge of +4 at pH 7). Similarly, [G10a]_3_-SHa has a length of 41 residues and an amidated C-terminus. This dimer has three free amino groups at the N-terminus and hydrophobic residues. It is cationic amphipathic and alpha-helical with D-Alanine residues (net charge of +6 at pH 7). Conversely, Jeff-[G10a]_2_-SHa has a length of 26 residues and an amidated C-terminus. This dimer has two free amino groups at the *N*-terminus and hydrophobic residues. It is cationic amphipathic (net charge of +2 at pH 7) and triple helix as secondary structure with D-Alanine residues. In addition, it has been conjugated to the lipophilic polymer Jeffamine trimer.

## 2. Materials and Methods

### 2.1. Reagents

All the reagents and chemicals employed for this experiment were 96–98% pure. Solid support was Fmoc protected Rink amide resin having loading capacity of 0.602 mmol/g. In addition, its mesh size was 200–300. All Fmoc amino acids and coupling reagents were acquired from Novabiochem and rest of the chemicals were purchased from Aldrich. All solvents were of HPLC grade. Chemical shifts are reported in ppm. Recycling preparative HPLC (LC-908) with SP-120-10 column (C-18) and solvent system in the ratio of ACN: H_2_O: TFA (60:40:0.08) were used for purification and UPLC to determine the percentage purity of all products.

### 2.2. Procedure for Synthesis of Peptide Dendrimers

#### 2.2.1. Synthesis of Temporin SHa Peptide and Its Analog [G10a]-SHa

Temporin SHa and its linear analog [G10a]-SHa were synthesized by using solid-phase peptides synthesis strategy (SPPS) as described by us earlier [14]. The UPLC profile, structure, synthetic scheme, ESI-MS data, ^1^H-NMR spectrum and NMR table of temporin-SHa and [G10a]-SHa are provided in the supporting information in Appendix A.

#### 2.2.2. Synthesis of Jeff-[G10a]_2_-SHa Conjugate

Tri-functional polymer (Fmoc)_2_ Jeffamine-acid linker [17,18] was synthesized from Jeffamine triamine T-403 containing repeating oxypropylene units and exists as a complex mixture of oligomers. In this procedure, one amino group of Jeffamine triamine T-403 was capped with succinic anhydride to produce a carboxyl group and two of its amino groups were confined by Fmoc to formulate trifunctional hydrophilic polymer as reported earlier [17,18] (Appendix A). Fmoc-protected Jeffamine linker (2.3 g, 3 equivalences) was loaded on the 1 g pre-swollen rink amide resin having loading capacity of 0.602 mmol/g. This step was repeated after 24 h to ensure the maximum coupling in the presence of coupling agents, oxymapure and DIC. After the removal of both Fmoc groups, [G10a]-SHa peptide was coupled in desired sequence using 6 equivalences of each amino acid, oxymapure and DIC, and at least 3 h of each coupling. For cleavage, TFA cocktail was used. The conjugated product was purified by RP-HPLC (Appendix A), and its purity was determined by UPLC (Appendix A). The structure of Jeffamine conjugate Jeff-[G10a]_2_-SHa was confirmed by various techniques, including ESI-MS mass (Appendix A) and ^1^H-NMR (Appendix A).

#### 2.2.3. Synthesis of [G10a]_2_-SHa Dendrimer

Peptide dendrimer [G10a]_2_-SHa was synthesized using Rink amide MBHA resin (Novabiochem, 0.51 mmol/g) in pre-swollen state in DCM and was swelled in 10 mL DMF (Appendix A). The resin was treated with 20% 4-methylpiperidine to take out Fmoc protecting group attached to the resin. Then, Fmoc-Lysine (Alloc)-OH (3 equiv.), Oxyma pure (0.65 g, 3 equiv.), and DIC (0.71 mL, 3 equiv.) were mixed in 10 mL of DCM and added in resin. The reaction mixture was left overnight to ensure maximum loading. Then, alloc group was removed with the help of palladium (0) catalyst using DCM: NMM: AcOH solvents and after this, Fmoc group was deprotected by deprotecting agent 20% 4-methylpiperidine in DMF for 25–35 min. Then, the desired sequence of peptide dendrimer was achieved by using Fmoc-protected amino acids (6 equiv. each), Oxyma pure (6 equiv.) and DIC (6 equiv.) dissolved in DCM. Reaction mixture was agitated for 4–6 h. Completion of reaction was checked by a ninhydrin colorimetric test. Elongation of the dimer peptide was continued, until both Fmoc groups on the last residues were removed. For cleavage, TFA cocktail was used comprising of TFA: EDT: Phenol: TIPS. The purity of dimer product was purified by RP-HPLC (Appendix A), and its purity was determined by UPLC (Appendix A). The structure of Jeffamine conjugate Jeff-[G10a]_2_-SHa was confirmed by ESI-MS mass (Appendix A) and ^1^H-NMR (Appendix A).

#### 2.2.4. Synthesis of [G10a]_3_-SHa Dendrimer

[G10a]_3_-SHa was synthesized using the same strategy of solid-phase peptides synthesis (SPPS) (Appendix A). For this, first, Fmoc-lysine (Alloc)-OH (3 equiv.), Oxyma pure (0.65 g, 3 equiv.), and DIC (0.71 mL, 3 equiv.) were mixed in 10 mL of DCM and added in resin. The reaction mixture was left overnight to ensure maximum loading. Afterwards, second Fmoc-protected lysine (Alloc)-OH (3 equiv.) was coupled in the same manner using oxymapure and DIC. Then, both alloc groups of both lysines were simultaneously removed with the help of palladium (0) catalyst using DCM: NMM: AcOH solvents and after this, Fmoc group of second lysine was deprotected by deprotecting agent 20% 4-methylpiperidine in DMF for 25–35 min. The desired sequence of trimer was acquired using Fmoc-protected amino acids (9 equiv each), Oxyma pure (9 equiv) and DIC (9 equiv) dissolved in DCM and reaction mixture was agitated for 6–8 h. Completion of reaction was checked by a ninhydrin colorimetric test. Elongation of the trimer peptide was continued until the Fmoc groups on the last residues were removed. For cleavage, TFA cocktail was used comprising of TFA: EDT: Phenol: TIPS. The trimer product was purified by RP-HPLC (Appendix A), and its purity was determined by UPLC (Appendix A). The structure of Jeffamine conjugate Jeff-[G10a]_2_-SHa was confirmed by ESI-MS mass (Appendix A) and ^1^H-NMR (Appendix A).

### 2.3. Peptide Purification

The peptides were purified by RP-HPLC on a LC-908 Japan Analytical Instrument eluted at a flow rate of 4 mL/min by 0.1% TFA in H_2_O/60% ACN. The purity of compounds was established by UPLC (Appendix A) and characterized by various spectral techniques (Appendix A).

### 2.4. CD Analysis

Far-ultraviolet CD spectra were recorded in a quartz cuvette of 10 mm path length. Temperature was set at 22 °C in CD_6_ spectropolarimeter. In addition, *d* (+)-10-camphorsulfonic acid was used to calibrate the output of instrument. All dendrimer peptides (1.0 mg/mL) were completely dissolved in SDS (Sodium dodecyl sulphate) in a concentrator of 15 µM and 20 mM SDS. Meanwhile, the wavelength of instrument was set between 185 to 260 nm and bandwidth of 2 nm. Four successive scans were run for each spectrum at the rate of 100 nm/min, while base line was acquired independently under the same condition, and secondary structures of all dendrimers were examined.

### 2.5. Stability and pH Study

The stability of dendrimers at different pH values and temperatures was also determined. For this, a 510 model digital pH meter (Oakton Eutech) with working electrode of glass material and a Ag/AgCl as reference electrode was used for the pH record. Dilute solutions of HCl and NaOH were added to adjust the pH of the solution from 2 to 12. Similarly, stability of monomer [G10a]-SHa and its three dendrimers was checked by heating the peptide dendrimers on different temperature ranges, i.e., room temperature, 35 °C, 40 °C, 50 °C and 60 °C.

### 2.6. Dynamic Light Scattering (DLS)

Mean hydrodynamic diameter, surface potential and PDI of dendrimeric peptides aggregates [G10a]_2_-SHa, [G10a]_3_-SHa, and Jeff-[G10a]_2_-SHa were evaluated in solution using zetasizer (DLS, Malvern system 4700, Malvern, UK). The dendrimeric solutions of peptides were transferred into transparent cuvette with care to remove bubble formation. The sample was then transferred to the cell holder and study was performed at 90° scattering angle at room temperature. The standard viscosity and refractive index were kept constant and kept at 1.0, 1.33 and 80.4 mPa, respectively.

### 2.7. Atomic Force Microscopy (AFM) Imaging

Morphology of branched peptides was studied with a digital instrument atomic force microscope (Agilent 5500, Chandler, AZ, USA) and used to image the dendrimers nanostructures. Mica slide was used for sample drop and it was air dried at ambient temperature. Furthermore, it was mounted on microscope. The whole analysis was performed in non-contact mode. Images collected were optimized and scans were taken at velocity of 1–5 µm/s and 512 × 512-line resolution.

### 2.8. Antimicrobial Assay

Antibacterial activity of aforementioned peptides was assessed through determination of the minimal inhibitory concentration (MIC). For this, two-fold serial dilutions of antimicrobial peptides in bacterial liquid media was used, as previously reported by National Committee of Clinical Laboratory Standards (NCCLS, 1997) [19,20,21]. For antibacterial activity, tests were conducted on different Gram-negative bacteriai.e., *Acinetobacter baumannii* (DSM 30007); *Escherichia coli* (ATCC 8739); *Enterobacter cloacae* (DSM 30054); *Helicobacter pylori* (ATCC 43504); *Klebsiella pneumoniae* (DSM 26371); *Pseudomonas aeruginosa* (ATCC 9027)) and gram-positive bacteria (*Bacillus subtilis* (ATCC 6633); *Enterococcus faecalis* (DSM 2570); *Enterococcus faecium* (DSM 20477); *Staphylococcus aureus* (ATCC 6538)). In addition, 3 mL of Mueller–Hinton (MH) broth was inoculated on specific agar plates by single colonies of the different bacterial strains. Incubation of tubes was performed at 37 °C for overnight while agitation was performed at 200 rpm. The very next day, 3 mL sterile MH was used to dilute the bacteria solution by 1/100. Moreover, incubation of these bacteria was carried out at 37 °C and 200 rpm to the point where these bacteria reached their log phase growth (optical density (OD) at 600nm around 0.6). Bacteria were then diluted in sterile medium to reach bacterial density around 10^5^ bacteria/mL. Then, 100 μL of this bacterial suspension was added per well of 96-well microplates (Greiner BioOne) and exposed to increasing concentrations of peptides (from 0 to 100 µM, 1:2 dilution). Plates were incubated at 37 °C for 18–24 h in aerobic (for *A. baumannii*, *E. cloacae*, *E. coli*, *P. aeruginosa*, *B. subtilis*, *S. aureus*) or micro-aerobic conditions (for *H. pylori, E. faecium* and *E. faecalis*). At the end of the incubation, OD_600nm_ was measured using microplate reader (Biotek, Synergy Mx). The MIC was defined as the lowest concentration of peptides that completely suppressed the visible growth of the bacteria. Experiments were performed in triplicate (*n* = 3).

### 2.9. Antiproliferative Assay

The antiproliferative effect of the peptides was determined by using human cells, as previously described [22,23,24,25,26,27]. Human cancer cells used in this experiment were ovarian cancer cells A2780 (obtained from ECACC), breast cancer cells MCF-7 (obtained from ATCC), liver cancer cells HepG2 (obtained from ATCC), lung cancer cells A549 (obtained from ATCC), pancreatic cancer cells MiaPaCa (obtained from ATCC), prostate cancer cells PC-3 (obtained from ATCC), and skin cancer cells MNT-1 (generous gift from the National Institute of Health (NIH) [28]. Normal human cells used were human lung fibroblasts IMR-90 (obtained from ATCC). All cells were cultured in DMEM which was supplemented with 10% fetal bovine serum (FBS), 1% L-glutamine, and 1% antibiotics (all from Invitrogen) at 37 °C in 5% CO_2_ incubator. For cytotoxicity assay, cells were maintained in 25 cm^2^ flasks. Cells were separated from flasks using trypsin-EDTA solution (from Thermofisher) and counted using Malassez chamber and diluted in culture medium before seeding into 96-well cell-culture plates (Greiner bio-one, Paris, France) at approximately 3000 cells/well. Cells were then treated with increasing concentration of peptides diluted in culture medium (1:2 serial dilution, from 100 µM to 0 µM). After 48 h, medium was aspirated and number of viable cells was colorimetrically determined using resazurin-based assay [19] following manufacturer’s instructions. The fluorescence intensity was measured using microplate reader (Biotek, Santa Clara, CA, USA, Synergy Mx) with an excitation wavelength of 530 nm and an emission wavelength of 590 nm. The fluorescence values were normalized by the controls (untreated cells) and expressed as percent of proliferation. The IC_50_ values of peptides (i.e., the concentrations causing 50% inhibition of cell proliferation) were determined using GraphPad^®^ Prism 7 software.

### 2.10. Hemolysis Assay

The ability of dendrimers and peptides to cause hemolysis was evaluated using human whole red blood cells [20,22]. Briefly, fresh human erythrocytes (obtained from Divbio Science Europe, NL) were washed 3 times by centrifugation at 800× *g* for 5 min with sterile phosphate buffer saline (PBS, pH 7.4). The washed erythrocytes were resuspended in complete culture media (DMEM supplemented with 10% FBS) to a final concentration of 8%. Human erythrocytes were then added into sterile 96-well microplates (150 µL per well) and were exposed to increasing concentrations of peptides or dendrimers. After 1 h incubation at 37 °C, plates were centrifuged at 800× *g* for 5 min. Then, 100 µL of supernatant were carefully transferred to a new 96-well microplate and absorbance was measured at 450 nm using microplate reader. Triton-X100 at 0.1% (*v*/*v*) was used as positive control giving 100% hemolysis. The HC_50_ values of peptides and dendrimers (i.e., the concentration of compound causing 50% of hemolysis) were calculated using GraphPad^®^ Prism 7 software.

## 3. Results

### 3.1. Amphiphilicity of Dendrimeric Peptides

The relative hydrophobicity of four compounds was quantified by UPLC. Different retention times of these dendrimers reflected the difference in their hydrophobicity. The net charge at neutral pH of the four compounds was calculated according to the free peptide calculator from the website of China Peptides Co., Ltd., Shanghai, China (https://www.bachem.com/knowledge-center/peptide-calculator/, accessed on 20 April 2022) and the retention times of dendrimers are in the order of 3.056, 3.057, 3.489, 4.019 and 9.61, respectively, thus indicating the hydrophobic order [G10a]_3_-SHa < Jeff-[G10a]_2_-SHa < [G10a]-SHa < [G10a]_2_-SHa (Table 2).

### 3.2. Secondary Structures Determination

Temporins, including SHa, although the unfold in water, are known to adopt a helical structure once in contact with lipids or molecules mimicking a lipidic environment, such as SDS [10]. The molecular structures of four dendrimeric peptides in the presence of SDS to mimic a lipid environment are shown in the Figure 2.

The folding and binding properties and secondary structure of these dendrimers were studied to see how branching effected the stabilities. Figure 2 shows that [G10a]_2_-SHa and [G10a]_3_-SHa demonstrated weak negative peaks at 221 nm and one intense negative peak at around 208 nm, respectively, which is characteristic of an alpha-helical structure. Meanwhile, a positive band was shown at around 195 to 197 nm, which shows the clear alpha-helical structure of these two dendrimers. In addition, Jeff-[G10a]_2_-SHa dendrimer manifested one weak minimum negative peak at around 208 nm and one intensive minimum band at 195 nm. This can also be concluded as a triple helix and this shift in peaks may be due to the conjugation of the peptide with the polymer molecule.

### 3.3. Hydrodynamic Diameter, Surface Charge, Polydispersity Index and Morphology of [G10a]_2_-SHa

The dynamic light scattering (DLS) technique was used to measure the average hydrodynamic diameter of about 149.5 nm of [G10a]_2_-SHa which is considerably less, as shown in Figure 3A. The zeta potential of [G10a]_2_-Sha was found to be −55.0 mV (Table 3), which is less than −30 mV and shows that [G10a]_2_-SHa particles are stable. Particle agglomeration was found to some extent due to interactions that usually induce conformational changes [29]. The polydispersity index (PDI) depicted the homogeneous dispersion of colloidal suspension. In our experiments, the PDI of [G10a]_2_-SHa peptide was 0.34 (Table 3), showing very little aggregation. The morphology of [G10a]_2_-SHa was found to be spherical with variation in sizes; the morphological character was evident by atomic force microscopy (Figure 3B).

### 3.4. Hydrodynamic Diameter, Surface Charge, Polydispersity Index and Morphology of [G10a]_3_-SHa

The dynamic light scattering (DLS) technique illustrated the average hydrodynamic diameter of about 46.13 nm of [G10a]_3_-SHa (Figure 4A), a range within 100 nm which could enhance uptake from 0.25 to 6 folds. The zeta potential of [G10a]_3_-SHa was found to be −71.2 mV, very far from +30 to −30 mV range (Table 3) and hence depicted the stability of the [G10a]_3_-SHa trimer peptide. It shows that [G10a]_3_-SHa particles are quite stable, even in dispersion form. Particle agglomeration is less likely to occur having high surface potential due to higher repulsion potential and thus it attains long-term stability [30]. Polydispersity index (PDI) showed a minute increase in the dispersion of colloidal suspension. In our experiments, the PDI of the [G10a]_3_-SHa peptide was 0.4, which shows the homogeneous dispersion of peptide composites (Table 3). The morphology of [G10a]_3_-SHa was found to be spherical and round shaped; the morphological character was evident by atomic force microscopy (Figure 4B).

### 3.5. Hydrodynamic Diameter, Surface Charge, Polydispersity Index and Morphology of Jeff-[G10a]_2_-SHa

The dynamic light scattering (DLS) technique depicts the average hydrodynamic diameter of about 174.1 nm of Jeff-[G10a]_2_-SHa (Figure 5A). The zeta potential of Jeff-[G10a]_2_-SHa was found to be −41.7 mV (Table 3). It showed the considerable stability of the peptide-polymer conjugated dendrimer. Particle agglomeration is less likely to occur having high surface potential due to higher repulsion potential and thus it attains long-term stability [30]. The polydispersity index (PDI) describes the uniform dispersion of colloidal suspension. In our experiments, the PDI of Jeff-[G10a]_2_-SHa peptide was 0.27, which shows the homogeneous dispersion of peptide composites (Table 3). The morphology of Jeff-[G10a]_2_-SHa was found to be somewhat spherical; the morphological character was clearly described by atomic force microscopy (Figure 5B). As investigations depicted, the results show that some aggregation tends to occur in peptides owing to their amphiphilic nature.

### 3.6. Stability and pH Study

Stability is an important parameter for the practical application of peptides. The change in particle size and its environment, including concentration, pH of the medium and temperature, also affect the stability of peptides. The goal of the present study is to assess the stability of dendrimeric peptides as a function of temperature and pH. The stability of monomer [G10a]-SHa and its three dendrimers was checked by heating the peptide dendrimers at different temperature ranges, i.e., room temperature, 35 °C, 40 °C, 50 °C and 60 °C, and their absorbance spectrum was recorded as shown in Figure 6. The results revealed that the monomer along with both dendrimers [G10a]_2_-SHa and [G10a]_3_-SHa showed considerable stability up to 60 °C, while the peptide conjugated with polymer Jeffamine Jeff-[G10a]_2_-SHa was stable at temperatures RT, 35 °C and 40 °C.

Similarly, the pH of dendrimers was adjusted by the addition of dilute HCl and NaOH solutions and a study of dendrimeric peptides was also completed at different pH ranges (Figure 7). The pH study results showed that the monomer [G10a]-SHa and its two dendrimers [G10a]_2_-SHa and [G10a]_3_-SHa were found to be significantly stable at pH values up to 10 and unstable at high pH values of 12 and 14. However, comparatively, Jeff-[G10a]_2_-SHa, a polymer-conjugated peptide dendrimer, showed considerably good stability at higher pH values but was not very stable at low pH ranges of pH 2 and pH 4. This may be due to the effect of the lipophilic polymer on the properties of peptides and the deprotonation of an OH functional group in a basic medium, which enhanced the stability of this conjugated peptide and reversed the trend as compared to other non-conjugated dendrimers, as shown in the Figure 7.

### 3.7. Antibacterial Activities of the Dendrimers

Temporin SHa has been shown to possess antibacterial activity. Similarly, its derivative [G10a]-SHa was tested and found to be active against *H. pylori* [16] and methicillin-resistant *Staphylococcus aureus* (MRSA) [15]. The antibacterial activity of [G10a]-SHa and its dendrimers was further tested and compared to SHa using various Gram-negative as well as Gram-positive bacterial strains, as shown in Table 4. Temporin SHa was found to be active on all the bacterial strains tested (MIC ranging from 1.56 to 50 µM). Its derivative [G10a]-SHa was found to be active on *A. baumannii*, *E. coli*, *H. pylori*, *K. pneumoniae*, *B. subtilis*, *E. faecalis*, *S. aureus* (with MIC ranging from 3.12 to 100 µM). However, on the other hand, contrarily to temporin, SHa[G10a]-SHa was not effective against *E. cloacae*, *P. aeruginosa*, and *E. faecium* (MIC > 100 µM). When comparing the activity of the dendrimers with [G10a]-SHa, it appeared that, depending on the strain tested, the dendrimers can have lower, similar or higher activity compared to [G10a]-SHa. For example, on *A. baumannii*, dendrimer [G10a]_3_-SHa was found to be four times more active than [G10a]-SHa, [G10a]_2_-SHa or Jeff-[G10a]_2_-SHa (MIC of 3.12 µM versus 12.5 µM). Similarly, on *E. coli*, [G10a]_3_-SHa was found to be four times more active than [G10a]-SHa (MIC of 25 µM versus 100 µM), whereas [G10a]_2_-SHa and Jeff-[G10a]_2_-SHa were found to be inactive (MIC > 100 µM). On *H. pylori*, [G10a]_2_-SHa, [G10a]_3_-SHa and Jeff-[G10a]_2_-SHa were all found to be less active than [G10a]-SHa ((MIC of 6.25, 25 and 25 µM versus 3.12 µM). On *K. pneumoniae*, although [G10a]-SHa gave a weak activity (MIC of 100 µM), none of the dendrimers were found to be active (MIC > 100 µM). Regarding Gram-positive bacteria, although the dendrimers gave the same MIC value as [G10a]-SHa on *B. subtilis* (i.e., MIC of 3.12 µM), differences were found using other strains. On *E. faecalis*, dendrimer [G10a]_2_-SHa was found to be twice as active than [G10a]-SHa or the other dendrimers (MIC of 12.5 µM versus 25 µM). On *E. faecium*, [G10a]-SHa was found to be inactive, but the dendrimers were active with MIC ranging from 6.25 to 12.5 µM. Finally, on *S. aureus*, [G10a]_2_-SHa was found to be active like [G10a]-SHa (MIC of 3.12 µM), but [G10a]_3_-SHa and Jeff-[G10a]_2_-SHa were less active (MIC of 6.25 µM).

### 3.8. Antiproliferative Activity of the Dendrimer Peptides

Giving the known anticancer effect of [G10a]-SHa [14], the antiproliferative activity of dendrimers based on [G10a]-SHa was evaluated through the determination of their IC_50_ on the proliferation of human normal or cancer cells (Figure 8, Table 5).

Temporin SHa was found to be inactive in terms of preventing cell proliferation with low to no effect up to 100 µM (IC_50_ > 100 µM), except on the breast cancer MCF-7 cells (IC_50_ of 20.36 ± 5.64 µM), as previously described [14]. The [G10a]-SHa analog displayed inhibitory activity against the proliferation of all human cancer-cell types tested (i.e., IC_50_ ranging from 13.04 ± 1.99 to 31.63 ± 1.56 µM, mean of 18.08 ± 6.68 µM) confirming previously published results [14]. Interestingly, [G10a]-SHa was found to be twice less active on normal human fibroblast IMR-90 cells, with an IC_50_ of 37.20 ± 2.88 µM, demonstrating its relative selectivity against cancer cells over normal ones. Dendrimers [G10a]_2_-SHa and [G10a]_3_-SHa exhibited IC_50_ on cancer cells ranging from 3.79 ± 1.61 to 14.31 ± 1.37 µM and 1.84 ± 0.17 to 6.94 ± 0.27 µM, respectively. With mean IC_50_ on human cancer cells of 6.85 ± 3.66 and 3.76 ± 1.87 µM, respectively, [G10a]_2_-SHa and [G10a]_3_-SHa are thus 2.6 and 4.8-times more active than [G10a]-SHa at inhibiting multiplying cancer cells. In addition, with IC_50_ values of 22.27 ± 0.40 and 13.94 ± 0.42 µM on normal human fibroblast (IMR-90 cells), dendrimers [G10a]_2_-SHa and [G10a]_3_-SHa were also found to be more selective than [G10a]-SHa at inhibiting the proliferation of human cancer cells over normal ones (3.2 and 3.7-fold selectivity on cancer over normal cells, respectively). Finally, dendrimer Jeff-[G10a]_2_-SHa was found to be less active than [G10a]-SHa or other dendrimers, with IC_50_ on cancer cells ranging from 33.92 ± 5.90 to > 100 µM and a mean activity of 52.13 ± 14.37 µM. In addition, with an IC_50_ of 66.24 ± 4.15 µM on normal human fibroblast cells, Jeff-[G10a]_2_-SHa was also found to be the least selective antiproliferative molecule, with a 1.2-fold difference between IC_50_ on human cancer and normal cells.

### 3.9. Hemolytic Activity of the Dendrimer Peptides

In order to evaluate the safety of the dendrimers, hemolysis assay was performed using human red blood cells (Figure 9). Whereas temporin SHa caused limited hemolysis (24.54 ± 1.88% at 100 µM, estimated HC_50_ of 266.20 ± 41.27 µM), the G10a-SHa peptide displayed a higher hemolytic effect (102.03 ± 6.18% at 100 µM, estimated HC_50_ of 43.66 ± 1.01 µM). Regarding dendrimers, [G10a]_2_-SHa and [G10a]_3_-SHa were, however, found to be highly hemolytic with complete hemolysis at 100 µM and an estimated HC_50_ of 11.55 ± 0.38 and 10.46 ± 0.36 µM, respectively. Interestingly, dendrimer Jeff-[G10a]_2_-SHa was shown to be the least hemolytic compound, with 24.01 ± 5.06% hemolysis at 100 µM and an estimated HC_50_ of 1926 ± 242 µM.

## 4. Discussion

Infection and cancers are major causes of human deaths, killing millions of people every year. In our previous studies, D-alanine substituted temporin-SHa analogs were recognized as being anticancer against myriad of human cancer cell lines (breast cancer, non-small-cell lung cancer, cervical cancer, and colorectal cancer) and antimicrobial (*H. pylori* and *S. aureus*) peptides [14,16]. Among different analogs tested in our earlier work, [G10a]-SHa was found to be the best candidate due to its potent antimicrobial and anticancer effect with good serum stability [15]. In the present study, the potential therapeutic applications of [G10a]-SHa and its newly synthesized dendrimers and Jeffamine conjugate were further evaluated. Dendrimers of [G10a]-SHa were designed in the form of di- and trimeric forms as [G10a]_2_-SHa and [G10a]_3_-SHa, respectively. Initially, a dimer of [G10a]-SHa was synthesized to study the effect of branching on the chemical and biological properties of [G10a]-SHa. In addition, a trimer of [G10a]-SHa was synthesized, where branching was carried out through two Lysine residues at the *C*-terminal. In addition, temporin peptide [G10a]_2_-SHa was conjugated with the trifunctional polymer Jeffamine triamine linker to produce a novel hybrid material Jeff-[G10a]_2_-SHa, which represents the diverse functionality of peptides and flexibility of polymers. This polymer was earlier used in the identification of anticancer peptidomimetics from one-bead-one compound combinatorial libraries. [17]. The conjugation of peptides with polymers results in stable confirmation and self-assembling into nano formulations, to exploit the responsiveness of peptides in the form of smart structures such as altered shape, structure and myriad properties [31,32,33]. The polymer chosen for conjugation was the Jeffamine T-403 polymer with enhanced hydrophilicity and porosity. Moreover, its biocompatibility, biodegradability and end-group modification with ease made it a better candidate for conjugation with peptides [17].

The dendrimeric peptides [G10a]_2_-SHa and [G10a]_3_-SHa and Jeff-[G10a]_2_-SHa were tested in term of stability and activities against different microbes and cancer cells.

In the temperature and pH stability study, [G10a]-SHa and its dimeric and trimeric forms showed higher stability up to 60 °C and pH values in the range of 2–14. However, Jeff-[G10a]_2_-SHa lost its stability at temperatures above 40 °C and pH values below 6.

In our previous studies, the antibacterial effect of [G10a]-SHa against various strains, including *H. pylori* and *S. aureus*, was reported [15,16]. The results showed that the newly synthesized dendrimeric analogs of G10a showed differing antibacterial activity against different bacteria. [G10a]_2_-SHa showed improved activity against *E. faecium* and *E. faecalis* with MICs of 6.25 and 12.5 µM, respectively. [G10a]_3_-SHa showed improved antibacterial activity against Gram-negative *E. coli* and *A. baumanii* with MICs of 25 and 3.12 µM, respectively. In addition, Jeff-[G10a]_2_-SHa did show antibacterial activity but with MICs values higher than [G10a]_2_-SHa.

Previously, [G10a]-SHa was found to be active against different human breast cancers, non-small cell lung cancers, and cervical cancers [14]. However, upon conjugating it with a breast-cancer targeting peptide, it selectively inhibited only human breast cancer, thereby losing its anticancer effects against other cancer cell lines [14]. In the current study, the antiproliferative activity of [G10a]_2_-SHa and [G10a]_3_-SHa were significantly increased, by at least two times, in all tested cancer cell lines as compared to [G10a]-SHa. In addition, although these analogs displayed an antiproliferative effect on normal human cells (human fibroblasts), their IC_50_ values were lower on cancer cells, demonstrating a net selectivity over tumor cells (an up to 3.7-fold difference for the more selective analog, i.e., [G10a]_3_-SHa). Dendrimer Jeff-[G10a]_2_-SHa was found to be the least active and the least selective analogs in terms of anticancer activity.

Finally, the safety of the dendrimers was measured and compared to the temporin SHa and its derivative G10a-SHa using hemolysis assay. Data obtained demonstrated that, although G10a, [G10a]_2_-SHa and [G10a]_3_-SHa display high hemolytic activity at concentrations close to the ones active on bacteria or cancer cells, dendrimer Jeff-[G10a]_2_-SHa possesses limited hemolytic effect, making it suitable for potential medical uses.

## 5. Conclusions

Overall, although the antibacterial and anticancer activities of dendrimers of [G10a]-SHa were significantly amplified as compared to monomer [G10a]-SHa, the high hemolytic activity of these dendrimers is limiting their use to topical treatment such as in the case of skin infection. On the contrary, a Jeffamine-based dendrimeric conjugate displays antibacterial and anticancer activities associated with low hemolytic action, making it potentially suitable for systemic treatment. Further studies will be required to confirm the therapeutic values of these newly synthesized peptide-dendrimers.

## Figures and Tables

**Figure 1 biomolecules-12-00770-f001:**
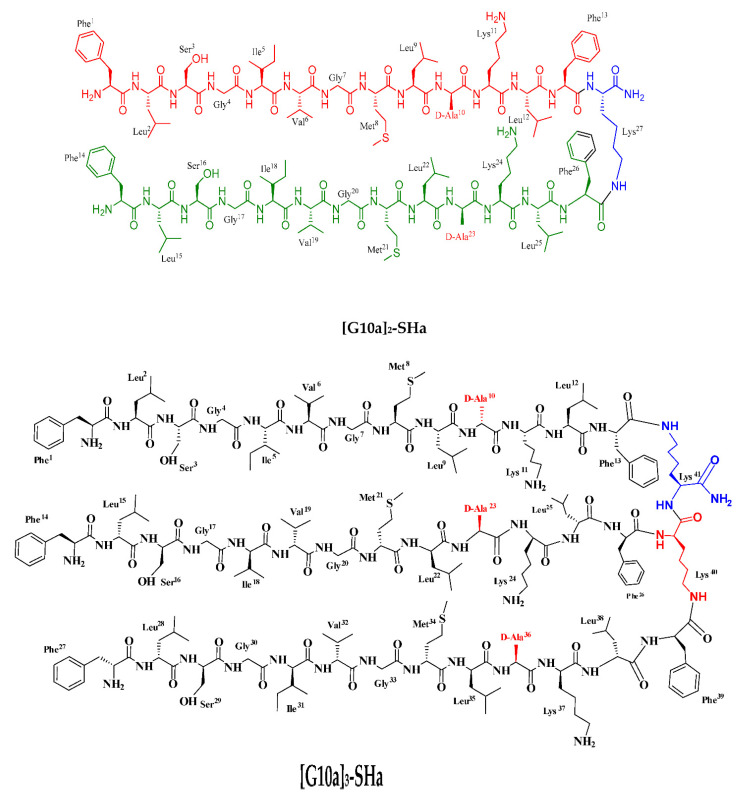
Structure of newly synthesized conjugate analogs of [G10a]-SHa after chemical modification.

**Figure 2 biomolecules-12-00770-f002:**
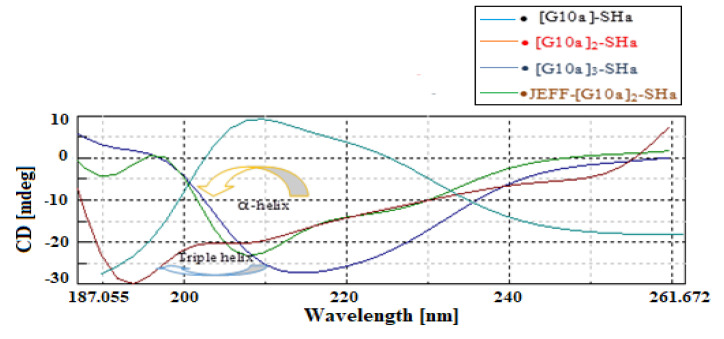
CD spectra of dendrimer peptides in 20 mM SDS solution.

**Figure 3 biomolecules-12-00770-f003:**
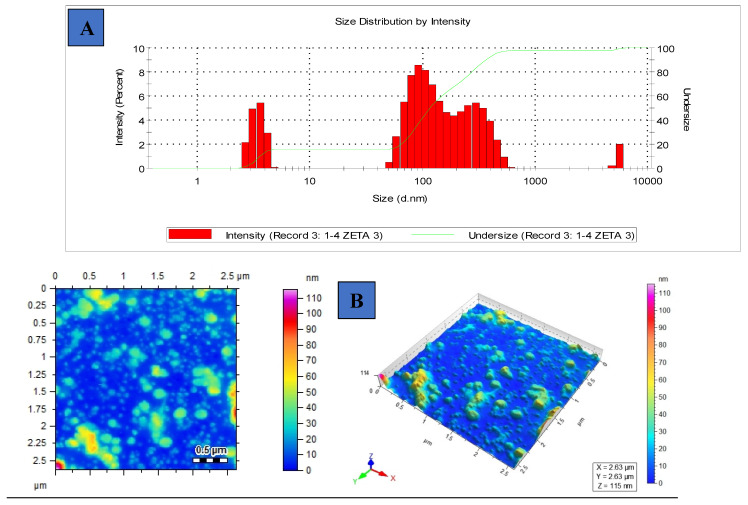
DLS intensity profile: (**A**) average size of [G10a]_2_-SHa using Zetasizer, (**B**) atomic force microscopic image of [G10a]_2_-SHa.

**Figure 4 biomolecules-12-00770-f004:**
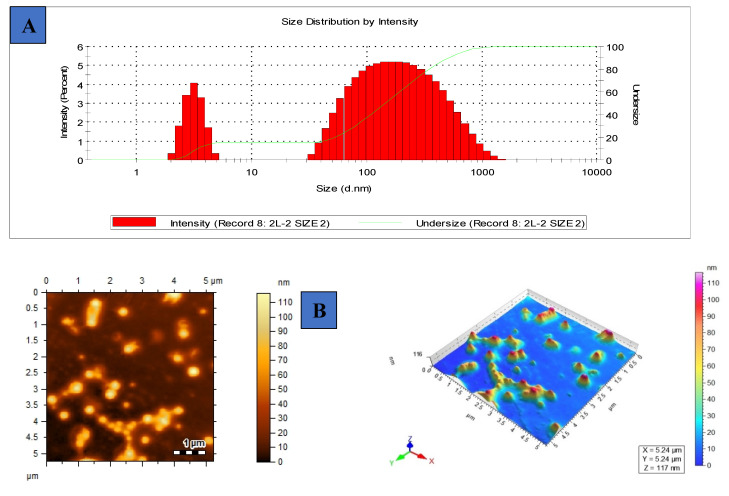
DLS intensity profile: (**A**) average size of [G10a]_3_-SHa using Zetasizer, (**B**) atomic force microscopic image of [G10a]_3_-SHa.

**Figure 5 biomolecules-12-00770-f005:**
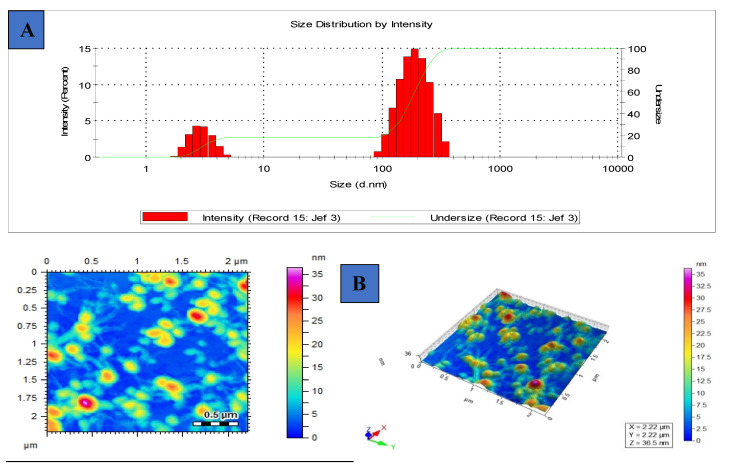
DLS intensity profile: (**A**) average size of Jeff-[G10a]_2_-SHa using Zetasizer, (**B**) atomic force microscopic image of Jeff-[G10a]_2_-SHa.

**Figure 6 biomolecules-12-00770-f006:**
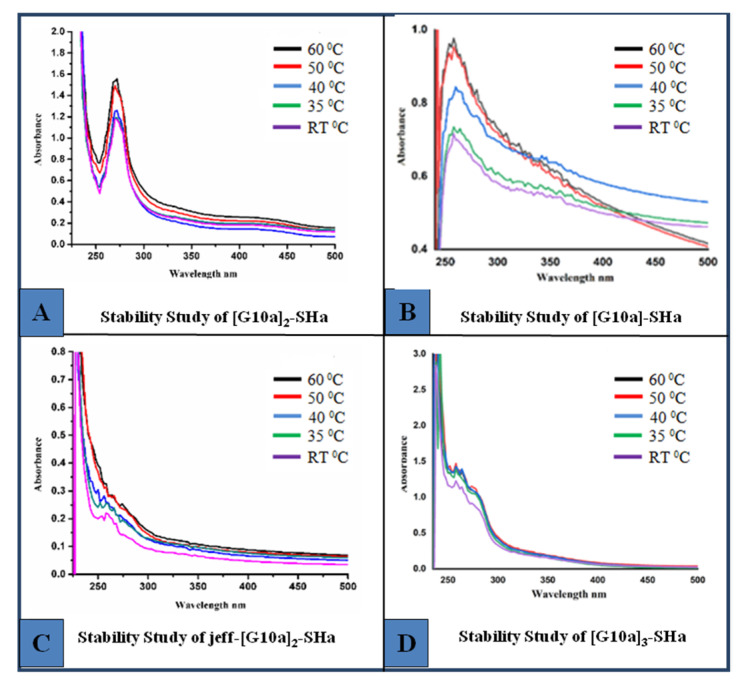
Stability studies of monomer and dendrimer peptides (**A**) [G10a]_2_-SHa (**B**) [G10a]-SHa (**C**) Jeff-[G10a]_2_-SHa (**D**) [G10a]_3_-SHa at different temperatures ranging from room temperature, 35 °C, 40 °C, 50 °C and 60°C.

**Figure 7 biomolecules-12-00770-f007:**
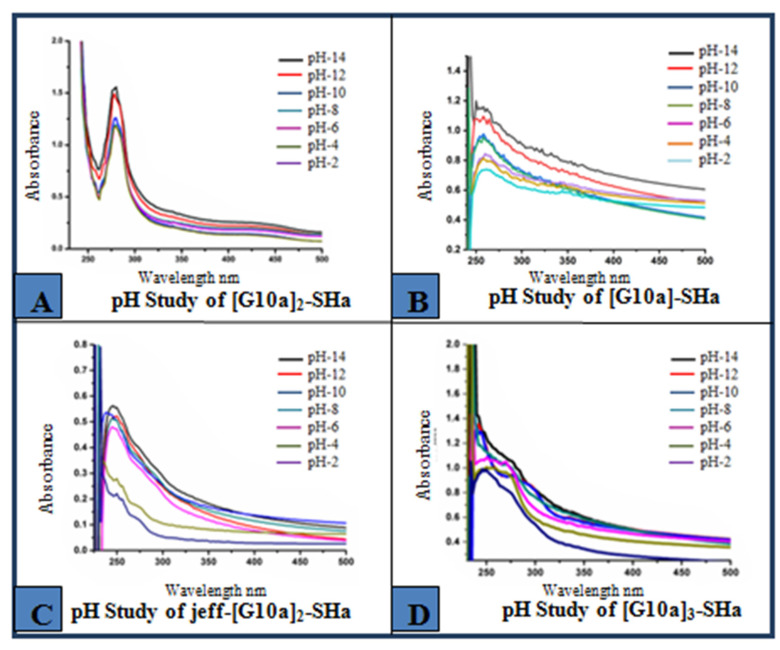
pH studies of monomer and dendrimer peptides (**A**) [G10a]_2_-SHa (**B**) [G10a]-SHa (**C**) Jeff-[G10a]_2_-SHa (**D**) [G10a]_3_-SHa at different ranges of pH from pH 2 to pH 14.

**Figure 8 biomolecules-12-00770-f008:**
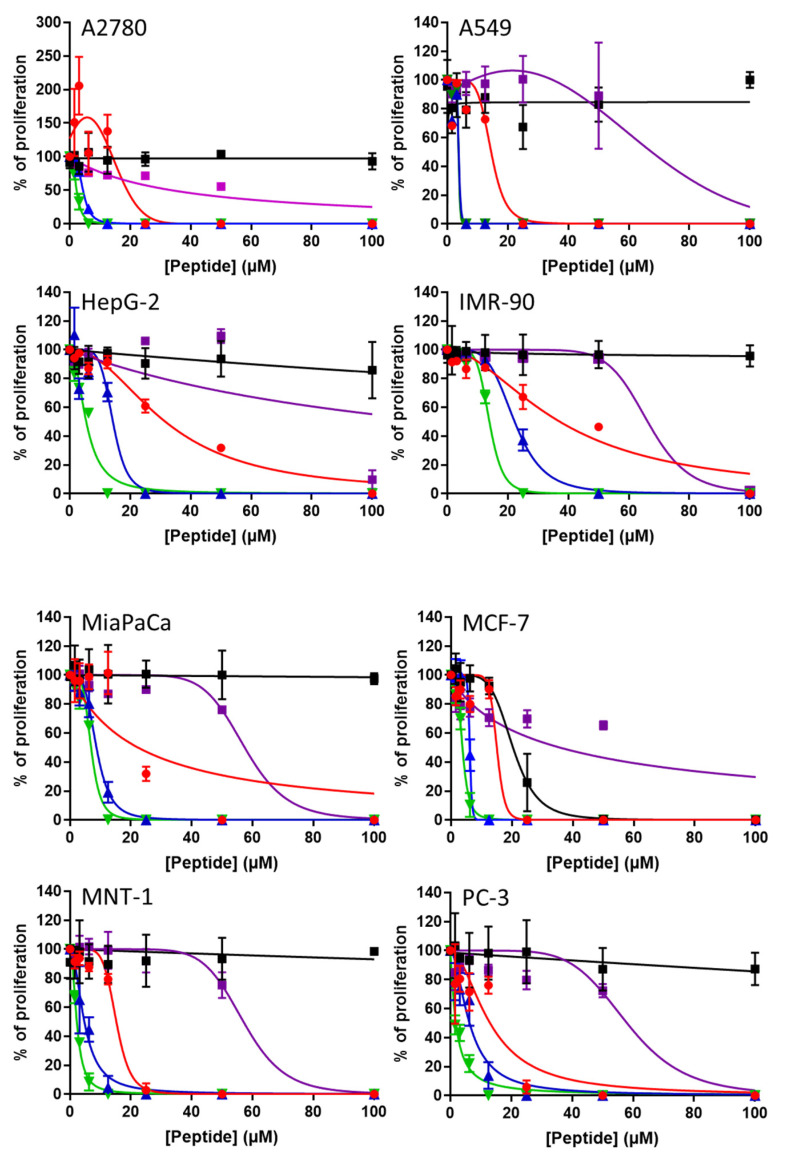
Antiproliferative effect of SHa, [G10a]-SHa and its dendrimers on human cancer cells. Proliferation of human cells over 48 h was measured in the presence of increasing concentrations of SHa (black squares), [G10a]-SHa (red circles), [G10a]_2_-SHa (blue triangles), [G10a]_3_-SHa (green inverted triangles) or Jeff-[G10a]_2_-SHa (purple squares). Proliferation was expressed as percentage of untreated control cells (means ± SD, *n* = 3). Curves were fitted using GraphPad^®^ Prism 7 software.

**Figure 9 biomolecules-12-00770-f009:**
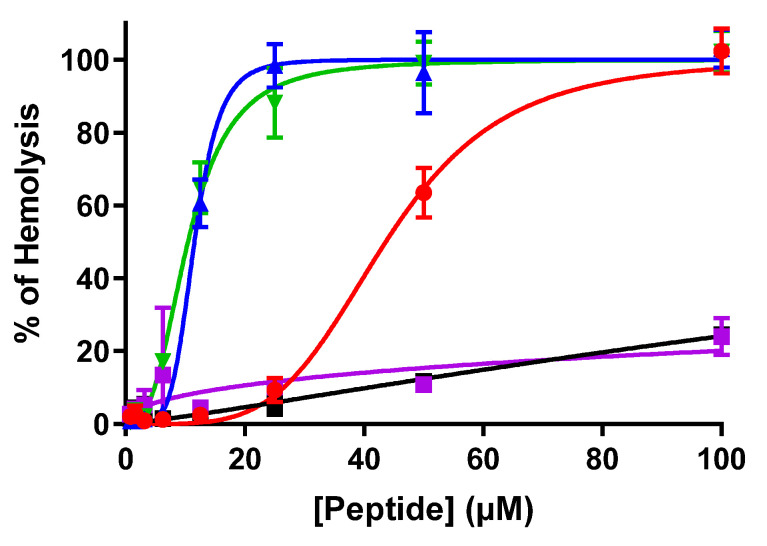
Hemolytic effect of SHa, [G10a]-SHa and its dendrimers on human red blood cells. Hemolysis was measured in the presence of increasing concentrations of SHa (black squares), [G10a]-SHa (red circles), [G10a]_2_-SHa (blue triangles), [G10a]_3_-SHa (green inverted triangles) or Jeff-[G10a]_2_-SHa (purple squares). Hemolysis was expressed in percentage of Triton X-100 0.1% being used as positive control giving 100% hemolysis (means ± SD, *n* = 3). Curves were fitted using GraphPad^®^ Prism 7 software.

**Table 1 biomolecules-12-00770-t001:** Sequence of linear analog of temporin-SHa peptide and its dimer, trimer and Jeff-conjugated dimer. Substitution of Gly is shown in red with D-Ala.

Peptides	Sequence
Temporin SHa	H-Phe^1^-Leu^2^-Ser^3^-Gly^4^-Ile^5^-Val^6^-Gly^7^-Met^8^-Leu^9^- Gly^10^-Lys^11^-Leu^12^-Phe^13^-NH_2_
[G10a] SHa	H-Phe^1^-Leu^2^-Ser^3^-Gly^4^-Ile^5^-Val^6^-Gly^7^-Met^8^-Leu^9^- D-Ala^10^-Lys^11^-Leu^12^-Phe^13^-NH_2_
[G10a]_2_ SHa	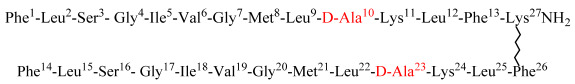
[G10a]_3_ SHa	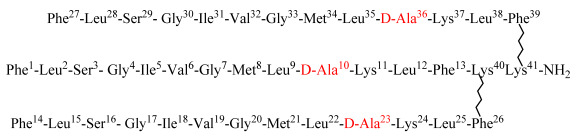
Jeff-[G10a]_2_ SHa conjugate	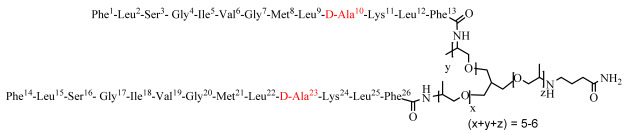

**Table 2 biomolecules-12-00770-t002:** Key physiochemical parameters of dendrimers showing molecular weights and retention time of the compounds obtained from UPLC.

Peptide Dendrimers	Observed Mass	Net Charge (pH 7.0)	*m*/*z*	Retention Time (min)	Overall Yield (%)
[G10a]-SHa	1393.82	+2	1394.96 [M + H]^+^	3.489	81.6
[G10a]_2_-SHa	2900.67	+4	1451.05 [M + 2H]^2+^	9.61	9.2
[G10a]_3_-SHa	4406.65	+6	1098.3 [M + 2H + Na]^3+^	3.056	9.8
Jeff-[G10a]_2_-SHa	~3294.44	+2 *	1106.64 [M + 3H]^3+^	3.057	4.8

* Net charge belongs to peptide moiety of dendrimer.

**Table 3 biomolecules-12-00770-t003:** Key nano-scale parameters of dendrimers of temporin SHa showing zeta size and potential along with polydispersity index values. Moreover, α-D values of dendrimers show their optical activity.

Peptide	Secondary Structure	Z-Average (d.nm)	Pdi Values	Zeta Potential (mV)	Optical Rotation
[G10a]-SHa	α-helical	63.25	0.628	−34.1	−178.46
[G10a]_2_-SHa	α-helical	149.5	0.39	−55.0	−387.69
[G10a]_3_-SHa	α-helical	46.13	0.462	−71.2	−229.1
Jeff-[G10a]_2_-SHa	Triple helical	174.1	0.277	−41.7	+213.0

**Table 4 biomolecules-12-00770-t004:** Antibacterial activity of dendrimers on various Gram-negative and Gram–positive bacteria. Antibacterial activity of the dendrimers is expressed as minimal inhibitory concentration (or MIC) in μM.

Peptides	Gram Negative						Gram Positive			
	*A. baumannii* (DSM 30007)	*E. cloacae* (DSM 30054)	*E. coli* (ATCC 8739)	*P. aeruginosa* (ATCC 9027)	*H. pylori* (ATCC 43504)	*K. pneumonia* (DSM 26371)	*B. subtilis* (ATCC 6633)	*E. faecalis* (DSM 2570)	*E. faecium* (DSM 20477)	*S. aureus* (ATCC 6538)
[G10a]-SHa	12.5	>100	100	>100	3.12	100	3.12	25	>100	3.12
[G10a]_2_-SHa	12.5	>100	>100	>100	6.25	>100	3.12	12.5	6.25	3.12
[G10a]_3_-SHa	3.12	>100	25	>100	25	>100	3.12	25	6.25	6.25
Jeff-[G10a]_2_-SHa	12.5	>100	>100	>100	25	>100	3.12	25	12.5	6.25
Temporin-SHa	6.25	50	50	50	3.12	50	1.56	12.5	6.25	3.12

**Table 5 biomolecules-12-00770-t005:** Antiproliferative activity of dendrimers on various human cells. Antiproliferative effect of the dendrimers was evaluated through the determination of their inhibitory concentration 50% (IC50), i.e., the concentration of dendrimers inhibiting 50% of the cell proliferation.

Peptides	Breast Cancer (MCF-7)	Liver Cancer (HepG-2)	Lung Cancer (A549)	Ovarian Cancer (A2780)	Pancreatic Cancer (MiaPaCa)	Prostate Cancer (PC-3)	Skin Cancer (MNT-1)	Human Normal Fibroblast (IMR-90)
I. [G10a]-SHa	15.05 ± 8.51	31.63 ± 1.56	14.52 ± 1.44	14.75 ± 7.35	22.36 ± 6.07	13.04 ± 1.99	15.25 ± 0.80	37.20 ± 2.88
II. [G10a]_2_-SHa	5.69 ± 1.46	14.31 ± 1.37	3.79 ± 1.61	4.42 ± 0.06	8.71 ± 0.35	6.31 ± 0.66	4.72 ± 0.36	22.27 ± 0.40
III. [G10a]_3_-SHa	3.77 ± 0.12	5.49 ± 0.37	3.74 ± 3.46	2.31 ± 0.08	6.94 ± 0.27	1.84 ± 0.17	2.24 ± 0.05	13.94 ± 0.42
IV. Jeff-[G10a]_2_-SHa	35.71 ± 8.50	>100	70.71 ± 5.39	33.92 ± 5.90	57.45 ± 6.03	57.99 ± 5.04	57.03 ± 6.09	66.24 ± 4.15
V. SHa	20.36 ± 5.64	>100	>100	>100	>100	>100	>100	>100

## Data Availability

All data are given in the main manuscript and Appendix A.

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
