# Peer review of "Design, Synthesis and Characterization of [G10a]-Temporin SHa Dendrimers as Dual Inhibitors of Cancer and Pathogenic Microbes"

_biomolecules, 2022, doi:10.3390/biom12060770_

Round 1

Reviewer 1 Report

In this paper, Khan et.al. have been investigated a temporin SHa named as [G10a]-SHa, its dendrimeric analogs [G10a]2-SHa and [G10a]3-SHa, and [G10a]2-SHa conjugated with polymer molecule i.e. jeff-[G10a]2-SH. They were using a different techniques to characterize stability and structure of those peptides. Additionally, antibacterial and antiproliferative biological activities were studied. 

General questions:

  1. Why do the Authors decide to study such extensive peptides? What is the biological purpose? Do the authors think they could be used as drugs? What about the enzymes that could digest them along the way?
  2. Why do the Authors use 20mM SDS in research? What about CMC (critical micelle concentration)?
  3. The Authors said that`The pH study result showed that the monomer [G10a]-SHa and its two dendrimer [G10a]2-SHa and [G10a]3-SHa were found much stable at pH up to 10 and instable at high pH of 12 and 14. But comparatively, jeff-[G10a]2-SHa, a polymer conjugated peptide dendrimer showed considerably good stability at higher pH but was poorly stable at low pH ranges of pH-2 and pH-4.` What is the biological significance of this?

In my opinion, it is a beautiful peptide chemistry, a very well prepared experimental part, but no clearly marked biological sense of the research. 

Author Response

Dear Reviewer,

please enclosed our answers:

"Why do the Authors decide to study such extensive peptides?"

Dendrimers of peptides have been proved to improve bioactivities compared to the peptides used to build them (Int. J. Mol. Sci. 2022, 23, 545.https://doi.org/10.3390/ijms23010545) and Chem. Rev. 2019, 119, 11391−11441. We previously published the anticancer and antimicrobial activity of G10a, a derivative of temporin SHa. Dendrimers of it were construct in order to see if such molecules possess higher activity compared to G10a

"What is the biological purpose?"

  1. As our group had previously synthesized the linear peptide G10a with considerably good anticancer and antibacterial activities (antibacterial activity of G10a was only tested on Helicobacter pylori, here we tested a large panel of strains). The biological purpose of synthesis of their dendrimers was to enhance the selectivity and bioactivities of these dendrimeric compounds compared to their corresponding linear presursor G[10a]. Peptide 106 (2018) 68–82 https://doi.org/10.1016/j.peptides.2018.07.002

"Do the authors think they could be used as drugs?"

  1. So far we have established the anticancer and antimicrobial potencies of these dendrimers of SHa. But the most active of them can be further developed as lead molecules. Various biomedical applications of peptide dendrimers such as active pharmaceutical agent and drug delivery carriers, in vaccine development and as diagnostic tools are reported. Based on the results of our study, we believe that such dendrimers have a real potential as drugs but we agree they still need to be improve and tested in vivo before to conclude that.

"What about the enzymes that could digest them along the way?"

We agree that proteolysis may theoritically reduce the concentrations of dendrimers in the body. But this will not be an issue in the case for exemple of topical treatment of skin infection. In addition, the literature shows that steric hindrance given by the bulky groups provides quite resistance to the proteases as mentioned in J. Mol. Sci. 2022, 23, 545. https://doi.org/10.3390/ijms23010545

1.      Why do the Authors use 20mM SDS in research?

2.      What about CMC (critical micelle concentration)?

  1. We used 20 mM SDS as the peptides dendrimers were not soluble below 20 mM concentration.

We did not determine the CMC as it was not related to our project.

"

The Authors said that`The pH study result showed that the monomer [G10a]-SHa and its two dendrimer [G10a]2-SHa and [G10a]3-SHa were found much stable at pH up to 10 and instable at high pH of 12 and 14. But comparatively, jeff-[G10a]2-SHa, a polymer conjugated peptide dendrimer showed considerably good stability at higher pH but was poorly stable at low pH ranges of pH-2 and pH-4.`What is the biological significance of this?"

We do not have any formal explanations regarding the pH sensitivity data. It is possible that the presence of Jeff in Jeff-G10a and its absence in other dendrimers constructs is reponsible by some way.

As far as pH stability studies of dendrimers are concerned, we performed the pH  studies of these dendrimers evaluate the extent of pH to which they are sensitive as stability of the antibiotics/drugs depends largely on the surrounding pH. The antibiotics/drugs which are sensitive towards selective pH are administered either in enteric coated formulation or else other stable formulation such as liposomes for eg actinomycin D which is already in market

Regards

Reviewer 2 Report

In the manuscript entitled "Design, Synthesis, and Characterization of [G10a]-Temporin SHa Dendrimers as Dual Inhibitors of Cancer and Pathogenic Microbes", Khan developed some dendrimers. The authors also characterized them and tested their anti-cancer and anti-microbial activities. The topic fits well the scope of Biomolecules. However, the reviewer feels it needs extensive amendments.
Here are the major comments:
(1) The anti-cancer and anti-microbial potencies of such dendrimers are quite low. What are the potential applications of such dendrimers?
(2) The dendrimers do not display much selectivity. The IC50 of transformed and non-transformed cells does not differ much. How to ensure such dendrimers are safe?
(3) It is well-known dendrimers cause hemolysis. The authors must test the hemolytic effects.
(4) What is the mechanism of action of such dendrimers?

Author Response

Dear Reviewer,

Please find enclosed our answers:

"The anti-cancer and anti-microbial potencies of such dendrimers are quite low. What are the potential applications of such dendrimers?"

We agree that the anticancer and antimicrobial activities of the dendrimers are most of the time similar to the ones of peptide G10a. But in particular cases, the dendrimeric constructs display improved activities. It is the case for particular bacterial species (such as Enterococcus faecium or Acinetobacter baumannii that are important human pathogen involved in nosocomial infection) where at least some dendrimers are more active than G10a. Similarly, some dendrimers showed improved activity and selectivity in anticancer testing, confirming such dendrimers (or improved version of them to come in the future) could represent interesting drug candidates.

"The dendrimers do not display much selectivity. The IC50 of transformed and non-transformed cells does not differ much. How to ensure such dendrimers are safe?"

We agree with the reviewer that the selectivity against cancer cells is not massive, but still important. Indeed, a low to nulle selectivity of action is also observed with antiproliferative drugs already used to treat cancer (explaining the side effect of such treatments). The safety was addressed using hemolysis assay as suggested in point 3 (see below).

"

It is well-known dendrimers cause hemolysis. The authors must test the hemolytic effects."

As suggested by the reviewer, we performed hemolysis assay and obtained results that have been added in the revised manuscript (in red in the text).

Moreover, high generation cationic dendrimers are attributed with the hemolytic activity  due to greater cationic charge as compared to their anionic counterparts. With the increase of cationic residues/charge, the hemolytic toxicity of dendrimers increases. That’s why, [G10a]3 showed such hemolysis due to presence of extensitve free cationic terminal groups.  This systematic increase in hemolytic data with branching as was cited in the literature that free cationic groups of dendrimers interact with the red blood cells and cause the hemolysis.

Drug Discovery Today  _ Volume 24, Number 5  _ May 2019 https://doi.org/10.1016/j.drudis.2019.02.014 and Biomolecules 2019, 9, 330, doi:10.3390/biom9080330 and Molecules 2009, 14, 3881-3905, doi:10.3390/molecules14103881

"

What is the mechanism of action of such dendrimers?"

Our previous publication showed that G10a is acting through membranolytic effect on Helicobacter pylori. Dendrimers of G10a certainly act the same way. 

regards

Round 2

Reviewer 2 Report

The manuscript has been improved. The reviewer has no objection to accept this manuscript.